# Coagulation measures after cardiac arrest (CMACA)

**Hyo Joon Kim**[1], **Kurz Michael**[2], **Jung Hee Wee**[3], **Joo Suk Oh**[4], **Won Young Kim**[5], **In Soo Cho**[6], **Mi Jin Lee**[7], **Dong Hun Lee**[8], **Yong Hwan Kim**[9], **Chun Song Youn**[1]*

1 Department of Emergency Medicine, Seoul St. Mary Hospital, College of Medicine, The Catholic University of Korea, Seoul, South Korea, 2 Department of Emergency Medicine, University of Alabama School of Medicine, Birmingham, Alabama, United States of America, 3 Department of Emergency Medicine, Yeouido St. Mary's Hospital, The Catholic University of Korea College of Medicine, Seoul, Korea, 4 Department of Emergency Medicine, Uijeongbu St. Mary's Hospital, The Catholic University of Korea College of Medicine, Uijeongbu, Korea, 5 Department of Emergency Medicine, Asan Medical Center, University of Ulsan College of Medicine, Seoul, Korea, 6 Department of Emergency Medicine, Hanil General Hospital, Korea Electric Power Medical Corporation, Seoul, Korea, 7 Department of Emergency Medicine, Kyungpook National University School of Medicine, Daegu, Korea, 8 Department of Emergency Medicine, Chonnam National University Medical School, Gwangju, Korea, 9 Department of Emergency Medicine, Samsung Changwon Hospital, Sungkyunkwan University School of Medicine, Changwon, Korea

* ycs1005@catholic.ac.kr

## Abstract

### Background

During cardiac arrest (CA) and after cardiopulmonary resuscitation, activation of blood coagulation and inadequate endogenous fibrinolysis occur. The aim of this study was to describe the time course of coagulation abnormalities after out-of-hospital CA (OHCA) and to examine the association with clinical outcomes in patients undergoing targeted temperature management (TTM) after OHCA.

### Methods

This prospective, multicenter, observational cohort study was performed in eight emergency departments in Korea between September 2018 and September 2019. Laboratory findings from hospital admission and 24 hours after return of spontaneous circulation (ROSC) were analyzed. The primary outcome was cerebral performance category (CPC) at discharge, and the secondary outcome was in-hospital mortality.

### Results

A total of 170 patients were included in this study. The lactic acid, prothrombin time (PT), activated partial thrombin time (aPTT), international normalized ratio (INR), and D-dimer levels were higher in patients with poor neurological outcomes at admission and 24 h after ROSC. The lactic acid and D-dimer levels decreased over time, while fibrinogen increased over time. PT, aPTT, and INR did not change over time. The PT at admission and D-dimer levels 24 h after ROSC were associated with neurological outcomes at hospital discharge. Coagulation-related factors were moderately correlated with the duration of time from collapse to ROSC.

**Data Availability Statement:** All relevant data are within the paper and its Supporting Information file

**Funding:** This research was supported by ZOLL Foundation. There was no additional external funding received for this study.

**Competing interests:** None of the authors has declared a conflict of interest.

## Conclusion

The time-dependent changes in coagulation-related factors are diverse. Among coagulation-related factors, PT at admission and D-dimer levels 24 h after ROSC were associated with poor neurological outcomes at hospital discharge in patients treated with TTM.

## Introduction

Despite advances in critical care, including targeted temperature management (TTM), out-of-hospital cardiac arrest (OHCA) still has high mortality and morbidity rates [1, 2]. Systemic inflammation and increased coagulation due to whole-body ischemia and reperfusion after cardiac arrest (CA) play an important role in hypoxic brain injury and multiple organ dysfunction. Once OHCA occurs, a lack of pulsatile blood flow facilitates rapid clot formation and subsequent return of spontaneous circulation (ROSC), distributing the clot burden throughout the vasculature and vital organs [3]. Therefore, optimal postarrest care requires careful management of the complex interaction between clot formation and its natural resolution.

The most common causes of OHCA are acute coronary syndrome (ACS) and pulmonary thromboembolism (PTE), which require treatment with systemic anticoagulation. Such therapy requires delicate titration that is problematic without a more complete understanding of coagulation dysfunction after CA. Moreover, TTM, which is considered the standard of care for postcardiac arrest syndrome (PCAS) patients, is known to impair coagulation and alter the metabolism of anticoagulation therapies. Coagulopathy in critically ill patients, such as those with sepsis and trauma, is known to be associated with poor outcomes [4–7]. Elevated D-dimer levels and high disseminated intravascular coagulation (DIC) scores are related to poor outcomes in PCAS patients [8–10]. Nevertheless, coagulation abnormalities after CA are not fully understood. The prognostic implication of coagulation-related factors measured repeatedly has not been studied in PCAS patients treated with TTM.

The aim of this study was to describe the time course of coagulation abnormalities after OHCA and to examine the association with clinical outcomes in patients undergoing TTM after OHCA.

## Methods

### Study design and setting

This prospective, multicenter, observational cohort study was performed in eight emergency departments of university-affiliated teaching hospitals in Korea between September 2018 and September 2019. Adult (over 19 years of age) comatose OHCA patients with ROSC treated with TTM irrespective of their initial rhythm and etiology of CA were enrolled. Patients were excluded if they had a cerebral performance category (CPC) > 3 before CA, if they had traumatic CA, if they had a do-not-resuscitate order (DNAR), if they had preexisting hereditary or induced coagulopathy, if they were members of a protected population (pregnant women and prisoners) or if they had a treatment plan including immediate use of systemic fibrinolysis. All patients were treated with TTM according to each hospital's protocol.

This study was approved by the institutional review board (IRB) of each participating hospital, including the IRB of Seoul St. Mary's Hospital (KC20RISI0722). Written informed consent was obtained from all patients' legal surrogates.

### Blood sample assays

Blood samples were collected two times using existing intravenous access. The first blood sample was collected within 60 minutes after ROSC, and the second was collected 24 hours after ROSC. The following were examined: prothrombin time (PT), activated partial thrombin time (aPTT), international normalized ratio (INR), lactic acid, D-dimer levels and fibrinogen. The manufacturers' normal ranges for the laboratory test levels were as follows: PT = 10.0~13.6 seconds (sec), aPTT = 22.8~34.6 sec, INR = 0.89~1.20, D-dimer = <0.08 mg/L, and fibrinogen = 160~350 mg/dl. All laboratory tests and interpretations were performed in accordance with the manufacturer's recommendations and standard operating procedures established by the Center.

### Outcome assessment

The primary outcome of this study was a poor neurological outcome at hospital discharge, defined as a CPC between 3 and 5. The CPC scale ranges from 1 to 5: 1 represents good cerebral performance or slight cerebral disability, 2 represents moderate disability or independent activities of daily life, 3 represents severe disability or dependence on others for daily support, 4 represents a comatose or vegetative state, and 5 represents death or brain death. The secondary outcome was in-hospital mortality.

### Statistical analysis

Continuous data are presented as the means ± standard deviations or medians and interquartile ranges (IQRs) as appropriate, and categorical variables are presented as counts and percentages. To compare differences in patient characteristics and outcomes, chi-square tests or Fisher's exact tests were performed for categorical variables, and Student's t test or Mann–Whitney U tests were performed for continuous variables. Univariate analysis was performed to determine the covariates for neurological outcomes at hospital discharge. Variables with a P value $\geq 0.157$ on univariate analysis were excluded from the multivariate logistic regression model. To examine the association between coagulation-related factors and neurological outcome at hospital discharge, multivariate logistic regression analyses with backward elimination were performed.

Statistical analyses were performed using SPSS 22.0 (Chicago, IL) and MedCalc 15.2.2 (MedCalc Software, Mariakerke, Belgium). P values $\leq 0.05$ were considered statistically significant.

## Results

Of the 202 patients who were observed during the study period, 32 were excluded from the final analysis because they died within 24 h after ROSC. Finally, 170 patients were included in this study. The mean age was 60.5 ± 15.7 years, and 126 (74.1%) patients were male. Forty-six (27.1%) patients had an initial shockable rhythm, 123 (72.4%) patients had poor neurological outcomes at hospital discharge, and 77 (45.3%) patients died prior to hospital discharge.

### Patient demographics

We divided the patients into two groups according to the neurological outcomes at hospital discharge. Table 1 shows the demographic and laboratory findings of the patients according to the neurological outcomes at discharge. Patients with poor neurological outcomes had a lower proportion of initial shockable rhythm, bystander CPR, and cardiac cause of arrest and had a longer time from collapse to the ROSC than those with good neurological outcomes.

**Table 1. Demographic findings according to neurological outcomes.**

|  | All | Good | Poor | *P* value |
|---|---|---|---|---|
| Number | 170 | 47 | 123 |  |
| Age | 60.8 ± 15.7 | 58.1 ± 13.6 | 61.8 ± 16.1 | 0.150 |
| Sex (male, %) | 126(74.1) | 36 (76.6) | 90 (73.2) | 0.648 |
| Witnessed cardiac arrest (%) | 99(58.2) | 33(70.2) | 6(54.1) | 0.057 |
| Bystander CPR (n, %) | 110(64.0) | 37(78.7) | 73(59.8) | 0.021 |
| Initial shockable rhythm (n, %) | 55(32.0) | 33(70.2) | 22(18.0) | <0.001 |
| Cardiac cause of arrest (n, %) | 105(61.0) | 40(85.1%) | 65(53.3) | <0.001 |
| Time from collapse to ROSC (min) [IQR] | 28.0 [17.00–39.00] | 17.0 [11.50–24.25] | 33.0 [22.0–43.25] | <0.001 |
| Total epinephrine dose (mg, IQR) | 2.0 [0–4.0] | 0.5 [0–2] | 2 [1–4.5] | <0.001 |
| Anticoagulation drug use during TTM |  |  |  |  |
| Heparin (n, %) | 42(24.7) | 16(34.0) | 26(21.1) | 0.081 |
| Nafamostat mesylate (n, %) | 10(5.9) | 1(2.1) | 9(7.3) | 0.198 |
| Laboratory findings at admission |  |  |  |  |
| Lactic acid (mmol/L) [IQR] | 9.7 [6.5–13.25] | 6.60 [4.00–10.25] | 10.25 [7.65–14.39] | <0.001 |
| Prothrombin time (sec) [IQR] | 13.60 [11.80–15.70] | 12.10 [11.10–13.45] | 14.10 [12.30–16.10] | <0.001 |
| Activated partial thromboplastin time (sec) [IQR] | 30.10 [25.48–37.90] | 25.70 [22.20–29.05] | 32.90 [26.50–45.20] | <0.001 |
| INR [IQR] | 1.18 [1.06–1.32] | 1.07 [0.98–1.14] | 1.23 [1.08–1.39] | <0.001 |
| Fibrinogen (mg/dl) [IQR] | 250.55 [190.50–323.75] | 262.0 [219.33–453.80] | 241.0 [178.0–362.0] | 0.183 |
| D-dimer (mg/L) [IQR] | 8.54 [3.01–20.32] | 3.54 [1.24–9.98] | 10.95 [4.14–32.33] | <0.001 |
| Laboratory findings 24h after ROSC |  |  |  |  |
| Lactic acid(mmol/L) [IQR] | 2.20 [1.30–3.85] | 1.20 [0.80–2.50] | 2.65 [1.43–4.05] | <0.001 |
| Prothrombin time (sec) [IQR] | 13.50 [12.20–16.40] | 12.60 [11.10–13.45] | 13.90 [12.70–17.55] | 0.001 |
| Activated PTT (sec) [IQR] | 31.40 [27.95–36.75] | 30.80 [28.50–36.20] | 33.80 [29.30–40.38] | 0.049 |
| INR [IQR] | 1.20 [1.09–1.39] | 1.13 [1.04–1.20] | 1.21 [1.09–1.46] | <0.001 |
| Fibrinogen (mg/dl) [IQR] | 346.80 [252.40–429.00] | 333.20 [252.40–431.00] | 354.20 [246.75–428.00] | 0.898 |
| D-dimer (mg/L) [IQR] | 3.48 [1.48–10.83] | 1.16 [0.46–3.65] | 3.96 [1.80–28.14] | <0.001 |

CPR = cardiopulmonary resuscitation, ROSC = return of spontaneous circulation, TTM = targeted temperature management, IQR = interquartile range

Anticoagulation treatment during TTM was not significantly different between the two groups. When comparing the laboratory findings between the two groups, lactic acid, PT, aPTT, INR, and D-dimer levels were higher in the patients with poor neurological outcomes at admission and 24 h after ROSC.

## Time course of coagulation-related factors

Lactic acid and D-dimer levels decreased over time, while fibrinogen increased over time. PT, aPTT, and INR did not change over time. When compared with the neurological outcomes, lactic acid and all coagulation-related factors except fibrinogen were different between the two groups. D-dimer levels and fibrinogen showed significant differences at admission and 24 h after ROSC in both those with good neurological outcomes and those with poor neurological outcomes. However, there was no difference in PT between the two groups, and aPTT differed only in those with good neurological outcomes at admission and 24 h after ROSC. Fig 1 shows the differences in coagulation-related factors in each group according to the neurological outcome.

## Logistic regression analysis

In the univariate analysis, witnessed CA, time from collapse to ROSC, cardiac cause of arrest, initial shockable rhythm, total epinephrine dose, lactic acid at admission, and lactic acid 24

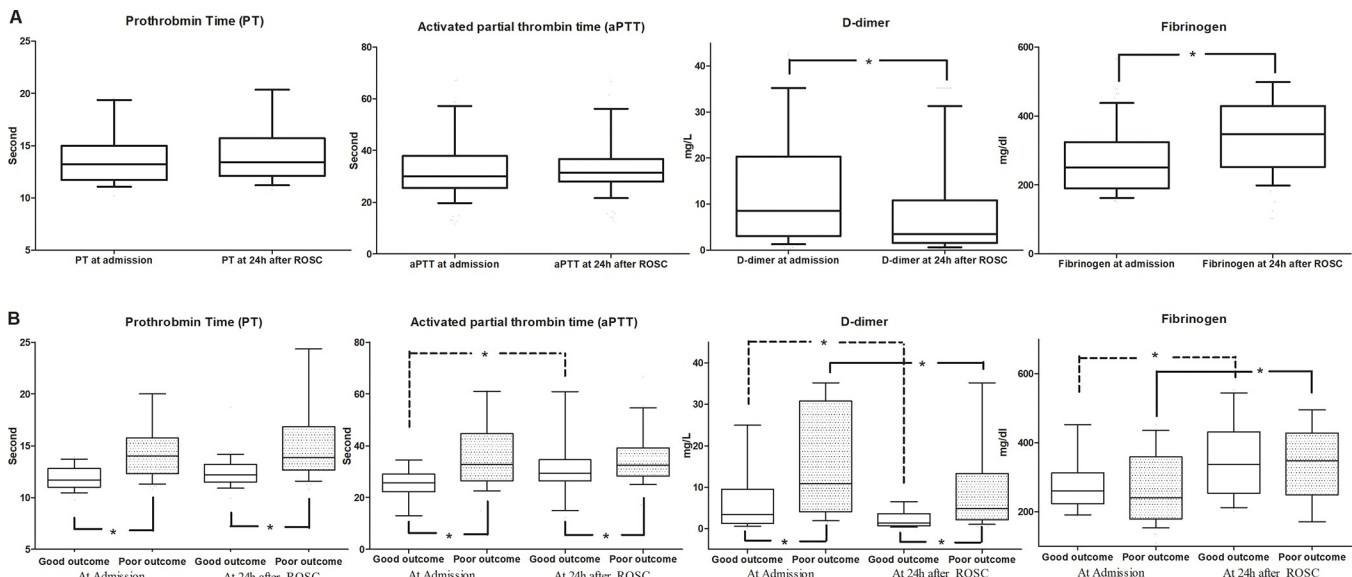

**Fig 1. Coagulation-related factors and outcome groups of the study patients.** The medians with interquartile ranges (box) and 10th-90th percentiles (whiskers) are presented. A represents coagulation-related factors between admission and 24 hours after ROSC. B represents coagulation-related factors according to neurological outcomes between admission and 24 hours after ROSC. (Differences between the groups are statistically significant at the p<0.05 level).

hours after ROSC showed statistically significant associations with neurological outcomes at hospital discharge. These were defined as covariates. After adjusting for covariates, PT at admission (aOR 1.652, 95% C.I. 1.095–2.493) and D-dimer levels 24 h after ROSC (aOR 1.209, 95% C.I. 1.043–1.400) showed an association with neurological outcomes at hospital discharge (Table 2).

For secondary outcomes, in the univariate analysis, time from collapse to ROSC, cardiac cause of arrest, initial shockable rhythm, total epinephrine dose, lactic acid at admission, and lactic acid 24 hours after ROSC showed statistically significant associations with in-hospital mortality. These were defined as covariates. After adjusting for covariates, fibrinogen at

**Table 2. Adjusted odds ratios of coagulation-related factors for predicting poor neurological outcome at hospital discharge.**

|  | aOR (95% CI) |
| --- | --- |
| PT at admission | 1.652 (1.095–2.493) |
| aPTT at admission | 1.009 (0.965–1.055) |
| INR at admission | 1.484 (0.655–3.365) |
| Fibrinogen at admission | 1.039 (0.993–1.087) |
| D-dimer at admission | 1.002 (0.997–1.006) |
| PT 24 h after ROSC | 1.396 (0.991–1.968) |
| aPTT 24 h after ROSC | 1.005 (0.975–1.035) |
| INR 24 h after ROSC | 1.185 (0.636–2.210) |
| Fibrinogen 24 h after ROSC | 1.001 (0.996–1.006) |
| D-dimer 24 h after ROSC | 1.209 (1.043–1.400) |

Adjusted by witnessed cardiac arrest, time from collapse to ROSC, cardiac cause of arrest, initial shockable rhythm, total epinephrine dose, lactic acid at admission, and lactic acid 24 hours after ROSC.

**Table 3. Adjusted odds ratios of coagulation-related factors for in-hospital mortality.**

| | aOR (95% C.I.) |
|---|---|
| PT at admission | 1.119 (0.991–1.265) |
| aPTT at admission | 1.011 (0.984–1.038) |
| INR at admission | 1.733 (0.831–3.612) |
| Fibrinogen at admission | 1.005 (1.001–1.008) |
| D-dimer at admission | 0.998 (0.981–1.015) |
| PT 24h after ROSC | 1.086 (0.990–1.191) |
| aPTT 24h after ROSC | 1.021 (0.994–1.049) |
| INR 24h after ROSC | 1.438 (0.839–2.465) |
| Fibrinogen 24h after ROSC | 1.001 (0.998–1.004) |
| D-dimer 24h after ROSC | 1.037 (0.998–1.077) |

Adjusted by time from collapse to ROSC, cardiac cause of arrest, initial shockable rhythm, total epinephrine dose, lactic acid at admission, and lactic acid 24 hours after ROSC.

admission (aOR 1.005, 95% C.I. 1.001–1.008) showed an association with in-hospital mortality (Table 3).

## Correlation between the duration of time from collapse to ROSC and coagulation-related factors

The duration of time from collapse to ROSC was moderately correlated with PT, aPTT, INR, fibrinogen, and D-dimer levels at admission (Spearman's rho = 0.471 ($p < 0.001$), 0.460 ($p < 0.001$), 0.498 ($p < 0.001$), 0.236 ($p = 0.002$), and 0.380 ($p < 0.001$), respectively). Lactic acid was also moderately correlated with the duration of time from collapse to ROSC (Spearman's rho = 0.364 ($p < 0.001$)).

## Prognostic value of coagulation-related factors

The PT at admission had an AUC of 0.806 (95% CI = 0.728–0.868, sensitivity: 68.32%, specificity: 85.71%, cutoff value: 13) for predicting poor neurological outcomes at hospital discharge. The D-dimer levels 24 h after ROSC had an AUC of 0.777 (95% CI = 0.707–0.837, sensitivity: 74.8%, specificity: 70.2%, cutoff value: 2.48) for predicting poor neurological outcomes at hospital discharge.

## Discussion

The main findings of this prospective, multicenter, observational cohort study regarding OHCA patients treated with TTM are as follows: first, lactic acid, PT, aPTT, INR, and D-dimer levels were higher in patients with poor neurological outcomes at admission and 24 h after ROSC. Second, lactic acid and D-dimer levels decreased over time, while fibrinogen increased over time. PT, aPTT, and INR did not change over time. Third, PT at admission and D-dimer levels 24 h after ROSC were associated with neurological outcomes at hospital discharge. Fourth, coagulation-related factors were moderately correlated with the duration of time from collapse to ROSC.

The cause of coagulopathy after CA is multifactorial. When OHCA causes a lack of pulsating blood flow, it promotes rapid clot formation in larger vessels and structures [11]. The individual mechanisms of hypotensive coagulopathy are known to be associated with poor outcomes in other critically ill patients, while coagulopathy resulting from CA is not well

understood. Additionally, global ischemic-reperfusion injury occurs after ROSC, so-called postcardiac arrest syndrome (PCAS), and it leads to massive dysfunction in the coagulation cascade [12–14]. Finally, TTM tempers the hypercoagulable state by slowing the breakdown of anticoagulant protein C (aPC) and altering platelet function [15].

Nevertheless, coagulofibrinolytic changes in PCAS patients treated with TTM are not fully understood. Endothelium, platelets, and coagulation factors are known as important components in the diagnosis of bleeding disorders and thus should be further studied in PCAS patients treated with TTM [16]. When an ischemic cardiovascular event occurs, breakdown of insulin-like growth factor binding protein occurs, increasing the production of CT-IGFBP-4. CT-IGFBP-4 is known to be associated with the early diagnosis and outcomes of ischemic cardiovascular events. In this study, D-dimer at admission was not associated with outcomes after CA. Therefore, it is necessary to study whether CT-IGFBP-4, an early marker, is also useful in PCAS patients treated with TTM [17].

D-dimer levels are a product of fibrinolysis, an indicator of thrombus fibrinolysis, and the severity of a hypercoagulable state [18]. When an intravascular thrombus is formed, D-dimer levels begin to be detected within 2 h, and the half-life time is approximately 6 h [19]. Once OHCA occurs, clots are formed due to no or low blood flow demonstrated by abnormally high levels of clotting byproducts, such as D-dimer levels [11]. In our study, most patients showed elevated D-dimer levels immediately after ROSC, reflecting a hypercoagulable state during CA. Once ROSC occurs, the clot burden accumulated during CA is distributed throughout the entire vasculature and vital organs. In response, the human body experiences a brief period of systemic fibrinolysis represented by the activation of aPC [20]. In our study, D-dimer levels decreased and fibrinogen increased over time, suggesting that systemic fibrinolysis occurred after ROSC.

In previous studies regarding CA, D-dimer levels at admission were found to be associated with outcomes [10, 21, 22]. However, in this study, D-dimer levels at admission were not associated with neurological outcomes. One possible explanation is that we included patients who received TTM after CA and survived more than 24 hours. Many previous studies have included patients with ROSC, not only those who have been treated with TTM [8, 10, 21, 22]. Another possible explanation is the cause of CA. The most common causes of OHCA are myocardial infarction and PTE, which can cause elevated D-dimer levels [23, 24]. Finally, D-dimer level elevation may reflect underlying hypercoagulability, pathologic fibrinolysis, and inflammatory processes [25–27]. Therefore, these multiple factors may explain why D-dimer levels at admission were not associated with poor neurological outcomes at hospital discharge in our study.

D-dimer levels at 24 h after ROSC were associated with poor neurological outcomes at hospital discharge in our study. D-dimer levels decreased over time in both the good and poor neurological outcome groups due to systemic fibrinolysis after ROSC [20, 28]. On the other hand, ischemic reperfusion injury causes intravascular thrombin formation by multiple pathways of inflammation, which may affect D-dimer levels in PCAS patients [29]. For this reason, patients with D-dimer levels that are still high at 24 h after ROSC were associated with poor neurological outcomes, possibly due to the influence of proinflammatory factors.

A positive correlation between the time from collapse to ROSC and D-dimer levels was reported previously. Asano et al. showed that D-dimer levels correlated with the duration of CA in witnessed OHCA, especially in OHCA due to cardiovascular causes [8]. Consistent with a previous study, D-dimer levels moderately correlated with the duration of time from collapse to ROSC in our study. In addition, other coagulation-related factors, such as PT, aPTT, INR and fibrinogen, were also moderately correlated with the time from collapse to ROSC. Therefore, coagulation-related factors may be possible surrogate markers of ischemic reperfusion injury after CA.

An increase in D-dimer levels at admission reflects a hypercoagulable state during CA, and a decrease in D-dimer levels at 24 hours reflects systemic fibrinolysis. The hypercoagulable state during CA leads to the no-reflow phenomenon, leading to hypoperfusion of vital organs, including the brain. A large-scale randomized study with tissue plasminogen activator during CA showed no evidence of a beneficial effect [30]. However, there have been no studies on the effects of anticoagulant agents in PCAS patients with ROSC.

Clinically, an increase in D-dimer levels at admission reflects a hypercoagulable state during CA, and a decrease over time reflects systemic fibrinolysis. An increase in the D-dimer level at admission was correlated with time from collapse to ROSC, and the D-dimer level at 24 h was associated with poor neurological outcomes after OHCA, which means that the results of systemic fibrinolysis may be related to the neurological outcome. Although we cannot recommend the use of anticoagulation agents, further studies are needed to suggest an optimal treatment strategy for coagulation abnormalities after CA.

This study has several limitations. Our study is an observational prospective study and may lead to a risk of selection bias and residual confounding. Second, neurological outcomes were evaluated at hospital discharge. The neurological outcomes 6 months after CA may be different. Third, the rates of unwitnessed arrest and an initial nonshockable rhythm were higher than those of other studies. Fourth, the effect of TTM on coagulopathy after CA was not evaluated.

## Conclusion

In conclusion, in terms of coagulation-related factors after CA, PT and aPTT did not differ between admission and 24 hours after ROSC. However, D-dimer levels decreased significantly, and fibrinogen increased significantly over time. PT at admission and D-dimer levels 24 h after ROSC were associated with poor neurological outcomes at hospital discharge in patients treated with TTM. Finally, coagulation-related factors were moderately correlated with the time from collapse to ROSC.

## Supporting information

**S1 Dataset.**
(XLSX)

## Author Contributions

**Conceptualization:** Hyo Joon Kim, Kurz Michael, Jung Hee Wee, Chun Song Youn.

**Data curation:** Hyo Joon Kim, Chun Song Youn.

**Formal analysis:** Hyo Joon Kim, Kurz Michael, Joo Suk Oh, Won Young Kim, In Soo Cho, Mi Jin Lee, Dong Hun Lee, Yong Hwan Kim, Chun Song Youn.

**Funding acquisition:** Hyo Joon Kim.

**Investigation:** Hyo Joon Kim, Joo Suk Oh, Won Young Kim, In Soo Cho, Mi Jin Lee, Dong Hun Lee, Yong Hwan Kim, Chun Song Youn.

**Methodology:** Chun Song Youn.

**Supervision:** Kurz Michael, Jung Hee Wee, Chun Song Youn.

**Writing – original draft:** Hyo Joon Kim, Chun Song Youn.

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
