## [Decision Letter · Decision Letter 0]

27 Oct 2022

PONE-D-22-13494Coagulation measures after cardiac arrest (CMACA)PLOS ONE

Dear Dr. Youn,

Thank you for submitting your manuscript to PLOS ONE. After careful consideration, we feel that it has merit but does not fully meet PLOS ONE’s publication criteria as it currently stands. Therefore, we invite you to submit a revised version of the manuscript that addresses the points raised during the review process.

we suggest the Authors to add a paragraph summarizing the main findings focusing on the clinical view. 

We look forward to receiving your revised manuscript.

Kind regards,

Chiara Lazzeri

Academic Editor

PLOS ONE

Journal Requirements:

2. Please provide the names of each of the institutional review boards which approved this study.

“This research was supported by ZOLL Foundation.”

Reviewers' comments:

Reviewer's Responses to Questions

**Comments to the Author**

1. Is the manuscript technically sound, and do the data support the conclusions?

Reviewer #1: Yes

2. Has the statistical analysis been performed appropriately and rigorously? 

Reviewer #1: Yes

3. Have the authors made all data underlying the findings in their manuscript fully available?

Reviewer #1: Yes

4. Is the manuscript presented in an intelligible fashion and written in standard English?

Reviewer #1: No

5. Review Comments to the Author

Reviewer #1: Congratulations to the author for this study.

1. New markers like CT-IGFBP-4 has been identified as predictive novel biomarker of Ischemic Cardiovascular events. https://www.ncbi.nlm.nih.gov/pmc/articles/PMC9420648/. Please look into this study and discuss this vital biomarker in your discussion.

2. Lately, endothelium, platelets, and coagulation factors as the three vital components has been identified for diagnosing bleeding disorders (https://www.ncbi.nlm.nih.gov/pmc/articles/PMC9440837/). Please discuss the probable role of the coagulation that measured in cardiac arrest.

3. The manuscript needs major English revision by a native speaker.

4. Authors also needs to give recommendation related to findings of this study.

6. PLOS authors have the option to publish the peer review history of their article (what does this mean?). If published, this will include your full peer review and any attached files.

Reviewer #1: No

---

## [Author Response · Author response to Decision Letter 0]

7 Nov 2022

We suggest the Authors to add a paragraph summarizing the main findings focusing on the clinical view. 

Thank you for your comments. I added the following sentences in the Discussion Section.

Clinically, an increase in D-dimer levels at admission reflects a hypercoagulable state during CA, and a decrease over time reflects systemic fibrinolysis. An increase in the D-dimer level at admission was correlated with time from collapse to ROSC, and the D-dimer level at 24 h was associated with poor neurological outcomes after OHCA, which means that the results of systemic fibrinolysis may be related to the neurological outcome. Although we cannot recommend the use of anticoagulation agents, further studies are needed to suggest an optimal treatment strategy for coagulation abnormalities after CA.

1. New markers like CT-IGFBP-4 has been identified as predictive novel biomarker of Ischemic Cardiovascular events. 

https://www.ncbi.nlm.nih.gov/pmc/articles/PMC9420648/. Please look into this study and discuss this vital biomarker in your discussion. 

Thank you for your comments. I added the following sentences in the Discussion Section.

When an ischemic cardiovascular event occurs, breakdown of insulin-like growth factor binding protein occurs, increasing the production of CT-IGFBP-4. CT-IGFBP-4 is known to be associated with the early diagnosis and outcomes of ischemic cardiovascular events. In this study, D-dimer at admission was not associated with outcomes after CA. Therefore, it is necessary to study whether CT-IGFBP-4, an early marker, is also useful in PCAS patients treated with TTM. [17]

17. Bhattarai A, Singh Sunar P, Shah S, Chamlagain R, Babu Pokhrel N, Khanal P, et al. CT-IGFBP-4 as a Predictive Novel Biomarker of Ischemic Cardiovascular Events and Mortality: A Systematic Review. Journal of Interventional Cardiology. 2022;2022:1-8. doi: 10.1155/2022/1816504.

2. Lately, endothelium, platelets, and coagulation factors as the three vital components has been identified for diagnosing bleeding disorders (https://www.ncbi.nlm.nih.gov/pmc/articles/PMC9440837/). Please discuss the probable role of the coagulation that measured in cardiac arrest.

Thank you for your comments. I added the following sentences in the Discussion Section.

Nevertheless, coagulofibrinolytic changes in PCAS patients treated with TTM are not fully understood. Endothelium, platelets, and coagulation factors are known as important components in the diagnosis of bleeding disorders and thus should be further studied in PCAS patients treated with TTM. [16]

16. Bhattarai A, Shah S, Bagherieh S, Mirmosayyeb O, Thapa S, Paudel S, et al. Endothelium, Platelets, and Coagulation Factors as the Three Vital Components for Diagnosing Bleeding Disorders: A Simplified Perspective with Clinical Relevance. International Journal of Clinical Practice. 2022;2022:1-10. doi: 10.1155/2022/5369001.

3. The manuscript needs major English revision by a native speaker.

Thank you for your comments. I received English proofreading by a native speaker at American Journal Experts. A proof of English proofreading is attached.

4. Authors also needs to give recommendation related to findings of this study.

Thank you for your comments. I added the following sentences in the Discussion Section.

An increase in D-dimer levels at admission reflects a hypercoagulable state during CA, and a decrease in D-dimer levels at 24 hours reflects systemic fibrinolysis. The hypercoagulable state during CA leads to the no-reflow phenomenon, leading to hypoperfusion of vital organs, including the brain. A large-scale randomized study with tissue plasminogen activator during CA showed no evidence of a beneficial effect. [31] However, there have been no studies on the effects of anticoagulant agents in PCAS patients with ROSC.

Clinically, an increase in D-dimer levels at admission reflects a hypercoagulable state during CA, and a decrease over time reflects systemic fibrinolysis. An increase in the D-dimer level at admission was correlated with time from collapse to ROSC, and the D-dimer level at 24 h was associated with poor neurological outcomes after OHCA, which means that the results of systemic fibrinolysis may be related to the neurological outcome. Although we cannot recommend the use of anticoagulation agents, further studies are needed to suggest an optimal treatment strategy for coagulation abnormalities after CA.

31. Abu-Laban RB, Christenson JM, Innes GD, Van Beek CA, Wanger KP, McKnight RD, et al. Tissue Plasminogen Activator in Cardiac Arrest with Pulseless Electrical Activity. New Engl J Med. 2002;346(20):1522-8. doi: 10.1056/nejmoa012885.

---

## [Editor Report · Decision Letter 1]

12 Dec 2022

Coagulation measures after cardiac arrest (CMACA)

PONE-D-22-13494R1

Dear Dr. Youn,

We’re pleased to inform you that your manuscript has been judged scientifically suitable for publication and will be formally accepted for publication once it meets all outstanding technical requirements.

Kind regards,

Chiara Lazzeri

Academic Editor

PLOS ONE
---

## [Editor Report · Acceptance letter]

28 Dec 2022

PONE-D-22-13494R1 

Coagulation measures after cardiac arrest (CMACA) 

Dear Dr. Youn:

I'm pleased to inform you that your manuscript has been deemed suitable for publication in PLOS ONE. Congratulations! Your manuscript is now with our production department. 

Kind regards, 

on behalf of

Dr. Chiara Lazzeri 

Academic Editor

PLOS ONE